# WHY PRE-TRAINING IS BENEFICIAL FOR DOWN-STREAM CLASSIFICATION TASKS?

## ABSTRACT

Pre-training has exhibited notable benefits to downstream tasks by boosting accuracy and speeding up convergence, but the exact reasons for these benefits still remain unclear. To this end, we propose to quantitatively and explicitly explain effects of pre-training on the downstream task from a novel game-theoretic view, which also sheds new light into the learning behavior of deep neural networks (DNNs). Specifically, we extract and quantify the knowledge encoded by the pre-trained model, and further track the changes of such knowledge during the fine-tuning process. Interestingly, we discover that only a small amount of pre-trained model's knowledge is preserved for the inference of downstream tasks. However, such preserved knowledge is very challenging for a model training from scratch to learn. Thus, with the help of this exclusively learned and useful knowledge, the model fine-tuned from pre-training usually achieves better performance than the model training from scratch. Besides, we discover that pre-training can guide the fine-tuned model to learn target knowledge for the downstream task more directly and quickly, which accounts for the faster convergence of the fine-tuned model. *The code will be released when the paper is accepted.*

## 1 INTRODUCTION

Pre-training is prevalent in nowadays deep learning, as it has brought great benefits to downstream tasks, including improving the accuracy (He et al., 2016; Devlin et al., 2019), boosting the robustness (Hendrycks et al., 2019), speeding up the convergence (Nguyen et al., 2023), and *etc*. Naturally, a fundamental question arises: **why pre-training is beneficial for downstream tasks?** Previous works have tried to answer this question from different perspectives. For example, Zan et al. (2022); Chen et al. (2023); Neyshabur et al. (2020) attributed the benefits of pre-training to a flat loss landscape. Erhan et al. (2010) concluded that the improved accuracy was a result of unsupervised pre-training acting as a regularizer.

Unlike above perspectives for explanations, we aim to present an in-depth analysis to answer the above question from a new perspective. That is, we quantify the knowledge encoded by the pre-trained model, and further analyze the effects of such knowledge on the downstream tasks. In this way, we can provide insightful and accurate explanations for the benefits brought by pre-training, which also sheds new light into the fine-tuning/learning behavior of DNNs.

To this end, we extract the knowledge encoded in the pre-trained model based on the interaction between different input variables (Ren et al., 2023a; Li & Zhang, 2023; Ren et al., 2024), because the DNN usually lets different input variables interact with each other to construct concepts for inference, rather than utilize each single variable for inference independently. As Fig. 1(a) shows, the DNN encodes the co-appearance relationship (interaction) between different image patches in $S = \{mouth, ear, eye\}$ of the input image $\boldsymbol{x}$ to form the *dog face* concept $S$ for inference. Only when all three patches in $S$ are all present, the interaction is activated and makes a numerical contribution $I(S|\boldsymbol{x})$ to the network output $y$. The absence/masking[1] of any image patch will deactivate the interaction, and the numerical contribution is removed, *i.e.*, $I(S|\boldsymbol{x}) = 0$.

More crucially, Ren et al. (2023a); Li & Zhang (2023) have empirically verified and Ren et al. (2024) have theoretically proven the **sparsity property** and the **universal-matching property** of interactions, *i.e., given an input sample x, a well-trained DNN usually encodes a small number of interactions between different input variables, and the network output y can be well explained as the*

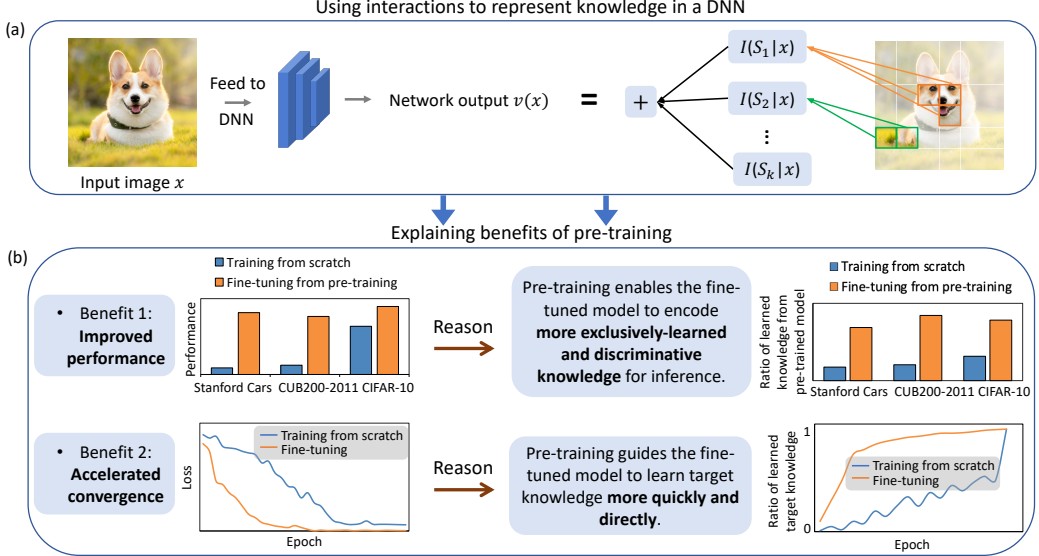

Figure 1: (a) We use the interaction between different input variables to represent knowledge encoded by a DNN, because the network output is proven to be well explained as the sum of numerical contributions $I(S|\boldsymbol{x})$ of interactions. (b) Explaining benefits of pre-training by analyzing effects of pre-trained model's knowledge on the downstream task.

numerical contributions of these interactions, $y = \sum_S I(S|\boldsymbol{x})$, as shown in Fig. 1(a). Thus, **these two properties mathematically enables us to take interactions as the knowledge encoded by the DNN for inference.** Apart from these two properties, the considerable discrimination power and high transferability across different models of interactions (Li & Zhang, 2023) also provide supports for the faithfulness of using interactions to represent knowledge encoded in a DNN. Please see Section 3.1 for detailed discussions.

In this way, we use interactions to precisely quantify and comprehensively analyze how pre-trained model's knowledge impacts the downstream classification task, so as to provide insightful explanations for two widely-acknowledged benefits of pre-training, *i.e.,* boosting the classification performance and speeding up the convergence. The following explanations may also guide some interesting directions on pre-training for future studies.

• **Quantifying explicit changes of pre-trained model's knowledge during the fine-tuning process.** We propose metrics to measure how pre-trained model's knowledge is discarded and preserved by the fine-tuned model for the inference of the downstream task, in order to provide comprehensive analyses for the benefits of pre-training. In experiments, we surprisingly discover that the fine-tuned model discards a considerable amount of pre-trained model's knowledge, especially extremely complex knowledge. In contrast, the fine-tuned model only preserves a modest amount of pre-trained model's knowledge that is discriminative for the inference of the downstream task.

• **Explaining the superior classification performance of the fine-tuned model.** We discover that only little preserved knowledge can be successfully learned by a model training from scratch merely using a small-scale downstream-task dataset, because the preserved knowledge from the pre-trained model is acquired from an extremely large-scale dataset. Thus, **pre-training makes the fine-tuned model encode more exclusively-learned and discriminative knowledge for inference**, which partially responses to the better accuracy of the fine-tuned model.

• **Explaining the accelerated convergence of the fine-tuned model.** Interestingly, we also observe that compared to the model training from scratch, **pre-training guides the fine-tuned model more quickly and directly to encode target knowledge used for the inference of the downstream task**, by proposing metrics to evaluate the learning speed of target knowledge and the stability of learning directions. Thus, this answers faster convergence of the fine-tuned model.

Contributions of this paper are summarized as follows. (1) We propose several theoretically verifiable metrics to quantify the knowledge encoded by the pre-trained model from a novel game-

theoretic view. (2) Based on the quantification of knowledge, we present an in-depth analysis to explain two benefits of pre-training. (3) Experimental results on various DNNs and datasets verify our explanations, which reveals new insights into pre-training.

## 2 RELATED WORK

**Explanation of pre-training.** Fine-tuning pre-trained models on downstream tasks to speed up convergence and boost performance has become a conventional practice in deep learning (He et al., 2016; Devlin et al., 2019; Hendrycks et al., 2019; Chen et al., 2023). Many works have attempted to analyze why pre-training is beneficial for downstream tasks from different perspectives. Specifically, Erhan et al. (2010) discovered that the unsupervised pre-training acted as a regularizer, which improved the generalization power of the DNN. Alternatively, a lot of studies explained the high accuracy (Zan et al., 2022; Neyshabur et al., 2020), the fast convergence speed in federated learning (Nguyen et al., 2023; Chen et al., 2023), and the reduced catastrophic forgetting in continual learning (Mehta et al., 2023) of the fine-tuned models from the perspective of a flat loss landscape. Additionally, Chen et al. (2024); Deng et al. (2023) explained the transferability of the pre-trained model to downstream tasks from the perspective of the feature space by performing the singular value decomposition. In comparison, we present a comprehensive analysis to systematically unveil the essential reasons behind different benefits of pre-training, by quantifying the explicit effects of pre-trained model's knowledge on the downstream task from a game-theoretic perspective.

**Using interactions to explain the DNN.** In recent years, employing game-theoretic interactions to explain DNNs has become a newly emerging direction. Specifically, Sundararajan et al. (2020); Tsai et al. (2023); Cheng et al. (2024) quantified interactions between different input variables to formulate the knowledge encoded by a DNN, whose faithfulness was further experimentally verified and theoretically ensured by (Li & Zhang, 2023; Ren et al., 2023a; 2024). Besides, a series of studies utilized the interaction to explain the representation capacity of DNNs, including the generalization power (Zhang et al., 2021; Yao et al., 2023; Zhou et al., 2024), adversarial robustness (Ren et al., 2021), adversarial transferability (Wang et al., 2021), the learning difficulty of interactions (Liu et al., 2023; Ren et al., 2023b), and the representation bottleneck (Deng et al., 2022). In comparison, this paper aims to provide insightful explanations for the benefits of pre-training to downstream tasks.

**Quantifying the knowledge encoded by the DNN.** So far, there does not exist a formal and widely accepted method to quantify the knowledge encoded by a DNN. A series of studies (Shwartz-Ziv & Tishby, 2017; Saxe et al., 2018; Higgins et al., 2017) employed the mutual information between input variables and the network output to quantify the knowledge in the DNN, but precisely measuring the mutual information was still significantly challenging (Kolchinsky et al., 2019). Besides, other studies employed human-annotated semantic concepts (Bau et al., 2017; Fong & Vedaldi, 2018) or automatically learned concepts (Chen et al., 2019) to explain the knowledge in the DNN, but these works could not quantify the exact changes of knowledge (*i.e.,* the preservation of task-relevant knowledge and the discarding of task-irrelevant knowledge) during the fine-tuning/training procedure. In comparison, we use theoretically verifiable interactions to represent knowledge in the DNN, which enables us to explicitly quantify the exact effects of pre-trained model's knowledge on the downstream task, so as to provide detailed explanations for the benefits of pre-training.

## 3 EXPLAINING WHY PRE-TRAINING IS BENEFICIAL FOR DOWNSTREAM TASKS

### 3.1 PRELIMINARIES: USING INTERACTIONS TO REPRESENT KNOWLEDGE IN DNNS

In this section, let us introduce the interaction metric, together with a set of interaction properties (Li & Zhang, 2023; Ren et al., 2023a; 2024) as convincing evidence for the faithfulness of interaction-based explanations, so as to provide a straightforward and concise way to understand why pre-training is beneficial for downstream tasks.

**Definition of interactions.** Given a DNN $v$ trained for the classification task and an input sample $\boldsymbol{x} = [x_1, x_2, \ldots, x_n]$ composed of $n$ input variables, let $N = \{1, 2, \ldots, n\}$ represent the indices of all $n$ variables. Let $v(\boldsymbol{x}) \in \mathbb{R}$ denote the scalar output of the DNN or a certain output dimension of the DNN, where people can apply different settings for $v(\boldsymbol{x})$. Here, we follow (Deng et al., 2022)

to set $v(\boldsymbol{x})$ as the confidence of classifying $\boldsymbol{x}$ to the ground-truth category $y^{\text{truth}}$ for multi-category classification tasks, as follows.

$$v(\boldsymbol{x}) = \log \frac{p(y = y^{\text{truth}}|\boldsymbol{x})}{1 - p(y = y^{\text{truth}}|\boldsymbol{x})}. \tag{1}$$

Then, the contribution of the interaction between a subset $S \subseteq N$ of input variables to the network output $v$ is calculated by the Harsanyi Dividend (Harsanyi, 1963), a typical metric in game theory, as follows.

$$I(S|\boldsymbol{x}) = \sum_{T \in S} (-1)^{|S|-|T|} \cdot v(\boldsymbol{x}_T), \tag{2}$$

where $\boldsymbol{x}_T$ denotes a masked input sample crafted by masking variables in $N \setminus T$ to baseline values[1] and keeping variables in $T$ unchanged. Let us take the sentence $\boldsymbol{x}$ ="he has a green thumb" as a toy example to understand equation 2. The DNN encodes the interaction between words in a subset $S = \{\text{green}, \text{thumb}\}$ with a numerical contribution $I(S)$ to push the DNN's inference towards the meaning of a *"good gardener."* This numerical contribution is computed as $I(S|\boldsymbol{x}) = v(\{\text{green}, \text{thumb}\}) - v(\{\text{green}\}) - v(\{\text{thumb}\}) + v(\boldsymbol{x}_\emptyset)$, where $\boldsymbol{x}_\emptyset$ denotes all words in $\boldsymbol{x}$ are masked.

**Understanding the physical meaning of interactions.** Each interaction with a numerical contribution $I(S|\boldsymbol{x})$ represents a collaboration (AND relationship) between input variables in a subset $S$. As in the aforementioned example, the co-appearance of two words in $S = \{\text{green}, \text{thumb}\}$ constructs a semantic concept of *"good gardener,"* and makes a numerical contribution $I(S|\boldsymbol{x})$ to the network output. The absence (masking) of any words in $S$ will inactivate this semantic concept and remove its corresponding interaction contribution, *i.e.*, $I(S|\boldsymbol{x}) = 0$.

**Quantifying the knowledge encoded by the DNN.** The proven *sparsity property* and *universal-matching property* of interactions enable us to use interactions to represent knowledge encoded by the DNN. Specifically, Ren et al. (2024) have proven that *under some common conditions[2], a well-trained DNN usually encodes very sparse interactions for inference*, which is also experimentally verified by Li & Zhang (2023); Zhou et al. (2024). In other words, although there exists $2^n$ different subsets[3] $S \subseteq N$ in total, only a small set $\Omega_{\text{salient}}$ of interactions make salient contributions to the network output, *i.e.*, $\Omega_{\text{salient}} = \{S \subseteq N, |I(S|\boldsymbol{x})| > \tau^4\}$, subject to $|\Omega_{\text{salient}}| \ll 2^n$. Whereas, a large number of interactions contribute negligibly $I(S|\boldsymbol{x}) \approx 0$ to the network output, which can be considered as noisy patterns. Thus, *the network output $v(\boldsymbol{x})$ can be well approximated by a small number of salient interactions*, *i.e.*,

$$v(\boldsymbol{x}) = \sum_{S \subseteq N} I(S|\boldsymbol{x}) \approx \sum_{S \in \Omega_{\text{salient}}} I(S|\boldsymbol{x}). \tag{3}$$

**Theorem 3.1** (**universal-matching property of interactions**). *Given an input sample $\boldsymbol{x}$, there are $2^n$ differently masked samples $\{\boldsymbol{x}_T | T \subseteq N\}$. Ren et al. (2024) have proven that network outputs $v(\boldsymbol{x}_T)$ on all $2^n$ masked samples $\boldsymbol{x}_T$ can be universally matched by a small number of salient interactions.*

$$v(\boldsymbol{x}_T) = \sum_{S \subseteq T} I(S|\boldsymbol{x}) \approx \sum_{S \subseteq T \& S \in \Omega_{salient}} I(S|\boldsymbol{x}). \tag{4}$$

Theorem 3.1 indicates we can use a small set of salient interactions to well explain the network output $v(\boldsymbol{x}_T)$ on anyone $\boldsymbol{x}_T$ of all $2^n$ masked samples. Thus, according to the Occam's Razor (Blumer et al., 1987), we can roughly consider **each salient interaction as the knowledge encoded by the DNN for inference**, rather than a mathematical trick with unclear physical meanings.

**Faithfulness of using interactions to represent the knowledge of the DNN.** Although nowadays there exist various methods to define/quantify the knowledge encoded by the DNN, *a set of theoretically proven and empirically verified interaction properties ensure the faithfulness of the interaction-based explanation*. Specifically, the *universal-matching property* in Theorem 3.1 and the *sparsity property* in equation 3 have mathematically guaranteed that interactions can faithfully

---

[1] We follow the widely-used setting in (Dabkowski & Gal, 2017) to set the baseline value of each variable as the mean value of this variable over all samples in image classification, and follow (Ren et al., 2023a) to set the baseline value of each word as a special token (*e.g.*, [MASK] token) in natural language processing.

[2] Please see Appendix B for the detailed introduction of common conditions.

[3] To reduce the computational cost, we select a relatively small number of input variables (image patches or words) to calculate interactions in experiments. Please see Appendix D.1 for details.

[4] $\tau$ is a small constant to select salient interactions, and we set $\tau = 0.05 \cdot \max_S |I(S|\boldsymbol{x})|$ in experiments.

explain the output of DNNs. Besides, Li & Zhang (2023) have experimentally verified the *transferability property* and the *discriminative property* of interactions. That is, interactions exhibit considerable transferability across samples and across models, and have remarkable discrimination power in classification tasks. Additionally, Ren et al. (2023a) have proven that interactions satisfy seven mathematical properties. Please see Appendix A for detailed discussions.

## 3.2 QUANTIFYING THE EFFECTS OF PRE-TRAINING ON DOWNSTREAM TASKS

Despite the ubiquitous utilization and great success of pre-trained models, it still remains mysterious why such models can help the fine-tuned model achieve superior classification performance and converge faster[5], compared to training from scratch. Thus, to systematically and precisely unveil the reasons behind these two benefits, we propose several metrics based on interactions to explicitly quantify the knowledge of the pre-trained model that is utilized for the inference of the downstream task, and further explain effects of such knowledge on the fine-tuning process. These explanations also provide some new insights into the learning/fine-tuning behavior of the DNN.

### 3.2.1 QUANTIFYING CHANGES OF PRE-TRAINED MODEL'S KNOWLEDGE DURING THE FINE-TUNING PROCESS

Explaining the precise effects of pre-training on downstream tasks still remains a significant challenge, because interactions (knowledge) directly extracted from the pre-trained model's output $v$ cannot be used for explanation. This is due to that the pre-trained model is usually trained on an extremely large-scale dataset with extensive training samples, whose network output often encodes a vast amount of diverse knowledge. Such knowledge can be further categorized into knowledge that can be used for inference of the downstream task (*e.g.,* some general and common knowledge), and knowledge that cannot be applicable to the downstream task (*e.g.,* knowledge only related to the inference of the pre-trained task). Thus, we need to extract and quantify the knowledge of the pre-trained model that is used for the inference of the downstream task for explanation, so as to ensure our explanation will not be affected by other irrelevant knowledge.

To this end, we employ the linear probing method (Alain & Bengio, 2016; Tenney et al., 2019; Liu et al., 2022; Chen et al., 2024), a commonly used technique, to extract pre-trained model's knowledge that is used for the downstream task. Specifically, given an input sample $\boldsymbol{x}$ and a pre-trained model, we freeze all its network parameters, and use the feature $f(\boldsymbol{x})$ of its penultimate layer (*i.e.,* the layer preceding the classifier of the pre-trained model) to train a new linear classifier $W^T f(\boldsymbol{x}) + b$ for the same downstream task as the fine-tuned model[6]. Then, we define the following function $v_{\text{pretrain}}$ to quantify the pre-trained model's knowledge used for the inference of the downstream task $I(S|\boldsymbol{x}, v_{\text{pretrain}})$, where $y_{\text{pretrain}}$ denotes the label predicted by the linear classifier.

$$v_{\text{pretrain}} = \log \frac{p(y_{\text{pretrain}} = y^{\text{truth}}|\boldsymbol{x})}{1 - p(y_{\text{pretrain}} = y^{\text{truth}}|\boldsymbol{x})}. \tag{5}$$

In this way, the classification score $v_{\text{pretrain}}$ enables us to provide a thorough insight into the effects of the pre-trained model on the downstream task, by quantifying the changes of its knowledge $I(S|\boldsymbol{x}, v_{\text{pretrain}})$ during the fine-tuning process. Specifically, we disentangle the knowledge $I(S|\boldsymbol{x}, v_{\text{pretrain}})$ into two components, including the knowledge preserved by the fine-tuned model for inference and the discarded knowledge. In this way, we define the preserved knowledge $K_{\text{preserve}}$ as the strength of the interaction shared by both the pre-trained model and the fine-tuned model. The discarded knowledge $K_{\text{discard}}$ is defined as the strength of the interaction that is encoded by the pre-trained model, but discarded by the fine-tuned model, as follows.

$$I(S|\boldsymbol{x}, v_{\text{pretrain}}) = \text{sign}(I(S|\boldsymbol{x}, v_{\text{pretrain}})) \cdot (K_{\text{preserve}}(S|\boldsymbol{x}) + K_{\text{discard}}(S|\boldsymbol{x})),$$
$$K_{\text{preserve}}(S|\boldsymbol{x}) = \mathbb{1}(\Gamma_{\text{pretrain}}^{\text{finetune}}(S|\boldsymbol{x}) > 0) \cdot \min(|I(S|\boldsymbol{x}, v_{\text{pretrain}})|, |I(S|\boldsymbol{x}, v_{\text{finetune}})|), \tag{6}$$
$$K_{\text{discard}}(S|\boldsymbol{x}) = |I(S|\boldsymbol{x}, v_{\text{pretrain}})| - K_{\text{preserve}}(S|\boldsymbol{x}),$$

where $\Gamma_{\text{pretrain}}^{\text{finetune}}(S|\boldsymbol{x}) = I(S|\boldsymbol{x}, v_{\text{pretrain}}) \cdot I(S|\boldsymbol{x}, v_{\text{finetune}})$ measures whether the interaction encoded by the pre-trained model $I(S|\boldsymbol{x}, v_{\text{pretrain}})$ and the interaction encoded by the fine-tuned model $I(S|\boldsymbol{x}, v_{\text{finetune}})$

---

[5]Experimental results in Appendix C verify that the fine-tuned model achieves higher classification accuracy and converges to a lower loss more quickly than the model training from scratch.

[6]Please see Appendix D.2 for the details of training the linear classifier.

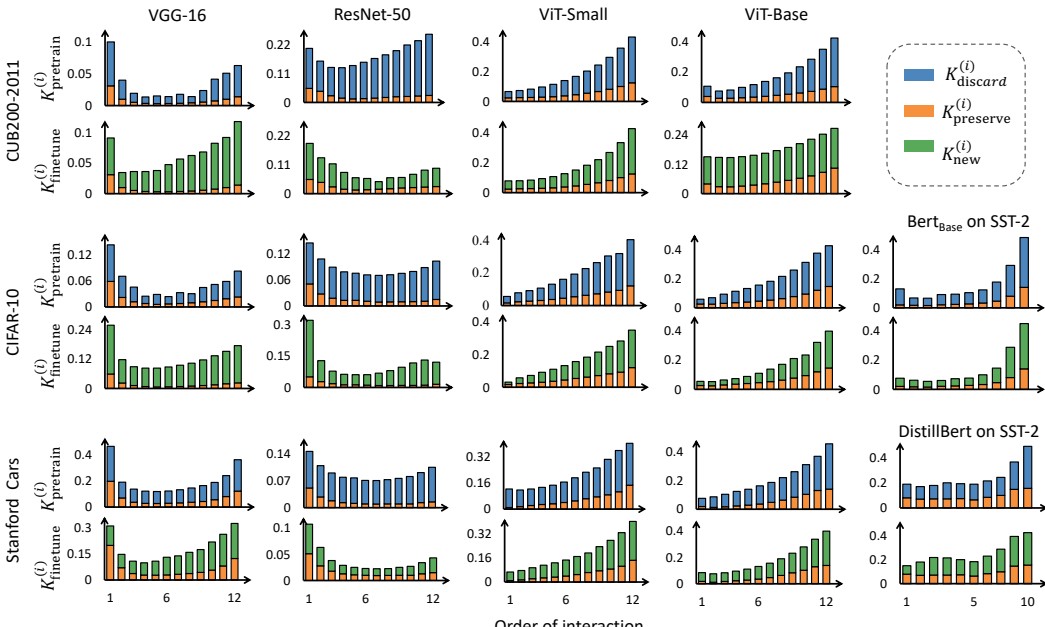

Figure 2: The preserved knowledge (interaction) $K_{\text{preserve}}^{(i)}$, the discarded knowledge $K_{\text{discard}}^{(i)}$, and the newly-learned knowledge $K_{\text{new}}^{(i)}$. For each subfigure, the total length of the blue bar and the orange bar equals to the knowledge encoded by the pre-trained model $K_{\text{pretrain}}^{(i)}$, and the length of the green bar and the orange bar equals to the knowledge encoded by the fine-tuned model $K_{\text{finetune}}^{(i)}$.

have the same effect. $v_{\text{finetune}}$ is calculated based on the fine-tuned model according to equation 1. $\mathbb{1}(\cdot)$ is the indicator function. If the condition inside is valid, $\mathbb{1}(\cdot)$ returns 1, and otherwise 0.

Similarly, we also disentangle the knowledge encoded by the fine-tuned model into two components, including the knowledge inherited from the pre-trained model $K_{\text{preserve}}(S|\boldsymbol{x})$, and new knowledge learned by the fine-tuned model itself to adapt the downstream task. Such a disentanglement helps us gain an insightful understanding of the fine-tuning behavior of the DNN, and also enables us to seek a deep exploration of the superior classification performance of the fine-tuned model in Section 3.2.2. Specifically, we define the knowledge $K_{\text{new}}(S|\boldsymbol{x})$ newly learned by the fine-tuned model as the strength of the interaction that is encoded by the fine-tuned model, but is not present in the pre-trained model.

$$I(S|\boldsymbol{x}, v_{\text{finetune}}) = \text{sign}(I(S|\boldsymbol{x}, v_{\text{finetune}})) \cdot (K_{\text{preserve}}(S|\boldsymbol{x}) + K_{\text{new}}(S|\boldsymbol{x})),$$
$$K_{\text{new}}(S|\boldsymbol{x}) = |I(S|\boldsymbol{x}, v_{\text{finetune}})| - K_{\text{preserve}}(S|\boldsymbol{x}). \tag{7}$$

**Experiments.** We conducted experiments to analyze changes of pre-trained model's knowledge during the fine-tuning process, in order to provide in-depth explanations for the effects of pre-training on downstream tasks. To this end, we employed off-the-shelf VGG-16 (Simonyan & Zisserman, 2015), ResNet-50 (He et al., 2016), ViT-Small, and ViT-Base (Dosovitskiy et al., 2021) pre-trained on the ImageNet-1K dataset (Russakovsky et al., 2015), and further fine-tuned these models on the CUB200-2011 (Wah et al., 2011), CIFAR-10 (Krizhevsky et al., 2009), and Stanford Cars (Krause et al., 2013) datasets for image classification, respectively. We also fine-tuned the pre-trained BERT$_{\text{BASE}}$ (Devlin et al., 2019) and DistillBERT (Sanh et al., 2019) models on the SST-2 (Socher et al., 2013) dataset for binary sentiment classification.

For a detailed explanation, we further quantified the preservation and the discarding of the knowledge of different complexities. The complexity of the knowledge was defined as the order of the interaction, *i.e.,* the number of input variables involved in the interaction, $complexity(S) = order(S) = |S|$. Thus, a high-order interaction denoted the interaction among a large number of input variables, which usually represented complex knowledge (interaction). In comparison, a low-order interaction among a small number of input variables was often referred to as simple and general knowledge.

Fig. 2 reports the average strength of the $i$-th order preserved interactions $K_{\text{preserve}}^{(i)} = \mathbb{E}_{\boldsymbol{x}}\mathbb{E}_{S \subseteq N, |S|=i}[K_{\text{preserve}}(S|\boldsymbol{x})]$, discarded interactions $K_{\text{discard}}^{(i)} = \mathbb{E}_{\boldsymbol{x}}\mathbb{E}_{S \subseteq N, |S|=i}[K_{\text{discard}}(S|\boldsymbol{x})]$, and newly-learned interactions $K_{\text{new}}^{(i)}$. Note that according to equation 6 and equation 7, the sum of $K_{\text{preserve}}^{(i)}$ and $K_{\text{discard}}^{(i)}$ equalled to the average strength of $i$-th order interactions encoded by the pre-trained model $K_{\text{pretrain}}^{(i)} = \mathbb{E}_{\boldsymbol{x}}\mathbb{E}_{S \subseteq N, |S|=i}[|I(S|\boldsymbol{x}, v_{\text{pretrain}})|]$, and the sum of $K_{\text{preserve}}^{(i)}$ and $K_{\text{new}}^{(i)}$ equalled to the average strength of $i$-th order interactions encoded by the fine-tuned model $K_{\text{finetune}}^{(i)} = \mathbb{E}_{\boldsymbol{x}}\mathbb{E}_{S \subseteq N, |S|=i}[|I(S|\boldsymbol{x}, v_{\text{finetune}})|]$. We discovered that even among different network architectures on different datasets, pre-training exhibits the similar effect on the downstream task, as follows.

• We surprisingly observed that **during the fine-tuning process, only a small amount of pre-trained model's knowledge was preserved for the inference of the downstream task, while a considerable amount of knowledge was discarded**, *i.e.,* the amount of the discarded knowledge was more than twice that of the preserved knowledge.

• Interestingly, we also discovered that **each fine-tuned model discarded more complex knowledge (reflected by high-order interactions) than simple and general knowledge (reflected by low-order interactions).** This indicated that complex knowledge encoded by the pre-trained model usually was not discriminative enough for the classification of the downstream task (*e.g.,* memorizing large-scale background patterns), thus the fine-tuned model discarded it, and re-learned discriminative knowledge for inference during the fine-tuning process.

• Correspondingly, **the fine-tuned model learned a large amount of new knowledge for the inference of the downstream task, especially complex knowledge.**

### 3.2.2 WHY THE FINE-TUNED MODEL CAN ACHIEVE SUPERIOR CLASSIFICATION PERFORMANCE?

Based on the quantification of pre-trained model's knowledge in the preceding section, here, we provide an insightful explanation for why pre-training can benefit the fine-tuned model in achieving superior classification performance[5]. Intuitively, we consider that compared to training from scratch, the fine-tuned model can preserve some discriminative knowledge from the pre-trained model, which is beneficial for making inference, such as classifying hard samples. This is due to that the preserved knowledge is usually acquired using a large-scale dataset with numerous training samples, thus it contains sufficiently discriminative information. More crucially, this knowledge preserved from the pre-trained model is very difficult to be learned by a DNN training from scratch merely using a small-scale downstream-task dataset. Thus, **pre-training makes the fine-tuned model encodes more exclusively-learned and discriminative knowledge than the model training from scratch for inference**, which accounts for the superior performance of the fine-tuned model.

To this end, we propose the following metric to examine whether the model training from scratch can only successfully learns a little preserved knowledge $K_{\text{preserve}}(S|\boldsymbol{x})$ for verification. Specifically, given a pre-trained model and its corresponding fine-tuned model, we train a randomly initialized DNN $v_{\text{random}}$ from scratch for the same downstream task, where we set it has the same network architecture as the fine-tuned model for fair comparisons. We quantify the ratio of pre-trained model's knowledge preserved by the fine-tuned model $K_{\text{preserve}}(S|\boldsymbol{x})$ that can be successfully learned by the model training from scratch, as follows.

$$ratio(S|\boldsymbol{x}) = \frac{\mathbb{1}(\Gamma_{\text{pretrain}}^{\text{random}}(S|\boldsymbol{x})) \cdot \min(|I(S|\boldsymbol{x}, v_{\text{random}})|, K_{\text{preserve}}(S|\boldsymbol{x}))}{K_{\text{preserve}}(S|\boldsymbol{x})}, \tag{8}$$

where $\Gamma_{\text{pretrain}}^{\text{random}}(S|\boldsymbol{x}) = I(S|\boldsymbol{x}, v_{\text{pretrain}}) \cdot I(S|\boldsymbol{x}, v_{\text{random}})$ measures whether interactions $I(S|\boldsymbol{x}, v_{\text{pretrain}})$ and $I(S|\boldsymbol{x}, v_{\text{random}})$ have the same effect to the network output. Only when interactions $I(S|\boldsymbol{x}, v_{\text{pretrain}})$, $I(S|\boldsymbol{x}, v_{\text{finetune}})$ and $I(S|\boldsymbol{x}, v_{\text{random}})$ have the same effect, the metric $ratio(S|x)$ is non-zero; Otherwise, $ratio(S|x) = 0$. A small value of $ratio(S|\boldsymbol{x})$ indicates that the model training from scratch can merely learn a little preserved knowledge $K_{\text{preserve}}(S|\boldsymbol{x})$.

**Experiments.** We conducted experiments to verify that the fine-tuned model encoded more exclusively-learned and discriminative knowledge than training from scratch. To this end, we trained randomly initialized VGG-16, ResNet-50, ViT-Small, and ViT-Base models on the CUB200-2011, CIFAR-10, and Stanford Cars datasets from scratch for image classification, respectively. We also trained randomly initialized BERT$_{\text{BASE}}$ and DistillBERT models on the SST-2 dataset from scratch for binary sentiment classification. Please see Appendix D.3 for more training details.

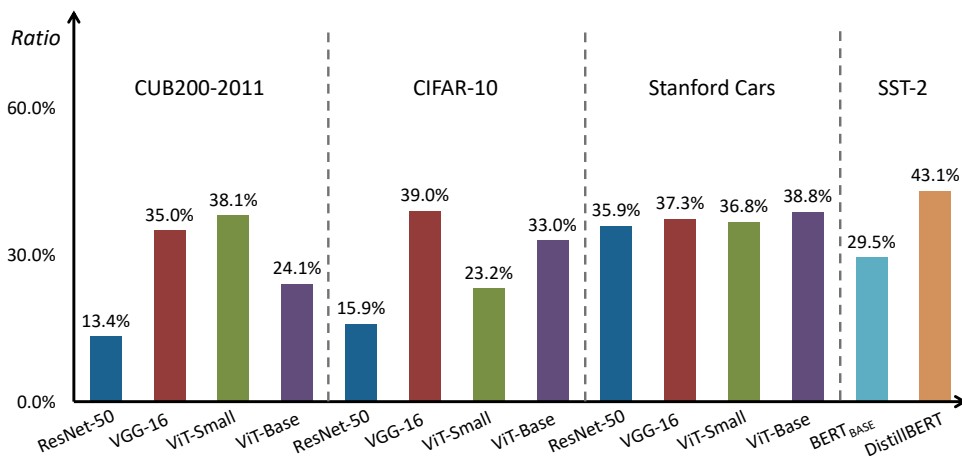

Figure 3: The ratio of the preserved knowledge that can be learned by the model training from scratch. This figure verifies that pre-training makes the fine-tuned model encodes more exclusively-learned and discriminative knowledge for inference than the model training from scratch, which responses to the superior performance of the fine-tuned model.

Fig 3 reports the average ratio of the preserved knowledge that the model training from scratch was able to learn, $Ratio = \mathbb{E}_{\boldsymbol{x}}\mathbb{E}_{S \subseteq N}[ratio(S|\boldsymbol{x})]$. We discovered that the average ratio for each DNN was very low, *i.e.,* ranging from 13% to 45%. This indicated that only a little preserved knowledge could be successfully learned by the model training from scratch, while most of it was extremely difficult to be acquired. Thus, compared to training from scratch, pre-training enabled the fine-tuned model to encode more exclusively-learned and discriminative knowledge for inference, resulting in its better performance.

### 3.2.3 WHY THE FINE-TUNED MODEL CONVERGES FASTER?

Apart from the improved performance, pre-training can also benefits the fine-tuned model in speeding up the convergence[5] (Hendrycks et al., 2019). In this section, we present an in-depth analysis to explain this benefit. Specifically, according to the information-bottleneck theory (Shwartz-Ziv & Tishby, 2017; Saxe et al., 2018), when training from scratch, the DNN usually tries to encode various knowledge in early epochs and discarding task-irrelevant knowledge in later epochs. In comparison, **pre-training guides the fine-tuned model to directly and quickly learn target knowledge, without temporarily modeling and discarding knowledge unrelated to the inference of the downstream task**, which is responsible for the faster convergence of the fine-tuned model.

Explicitly speaking, whether or not a DNN can quickly and directly learn target knowledge can be analyzed as whether the amount of learned target knowledge increases fast and stably along with the epoch number, respectively, where we define the target knowledge as the interaction encoded by the finally-learned DNN. To this end, we propose the following metrics to examine whether the fine-tuned model encodes target knowledge more directly and quickly for verification. Specifically, let the vectors $\boldsymbol{I}_{\text{finetune},e}(\boldsymbol{x}) = [I(S_1|\boldsymbol{x}, v_{\text{finetune},e}), I(S_2|\boldsymbol{x}, v_{\text{finetune},e}), \cdots, I(S_d|\boldsymbol{x}, v_{\text{finetune},e})] \in \mathbb{R}^d$ and $\boldsymbol{I}_{\text{finetune},E}(\boldsymbol{x})$ represent the distribution of all interactions encoded by the model fine-tuned after $e$ epochs and $E$ epochs, respectively, where $E$ denotes the total epoch number. Accordingly, the vector $\boldsymbol{I}_{\text{random},E}(\boldsymbol{x})$ and the vector $\boldsymbol{I}_{\text{random},E}(\boldsymbol{x})$ represent the distribution of all interaction encoded by the model training from scratch after $e'$ epochs and $E'$ epochs, respectively. Then, we calculate the Jaccard similarity between interactions encoded by the DNN learned after certain epochs and those encoded by the finally-learned DNN.

$$Jaccard_{\text{finetune}} = \mathbb{E}_{\boldsymbol{x}} \left[ \| \min(\tilde{\boldsymbol{I}}_{\text{finetune},e}(\boldsymbol{x}), \tilde{\boldsymbol{I}}_{\text{finetune},E}(\boldsymbol{x})) \|_1 / \| \max(\tilde{\boldsymbol{I}}_{\text{finetune},e}(\boldsymbol{x}), \tilde{\boldsymbol{I}}_{\text{finetune},E}(\boldsymbol{x})) \|_1 \right],$$
$$Jaccard_{\text{random}} = \mathbb{E}_{\boldsymbol{x}} \left[ \| \min(\tilde{\boldsymbol{I}}_{\text{random},e'}(\boldsymbol{x}), \tilde{\boldsymbol{I}}_{\text{random},E'}(\boldsymbol{x})) \|_1 / \| \max(\tilde{\boldsymbol{I}}_{\text{random},e'}(\boldsymbol{x}), \tilde{\boldsymbol{I}}_{\text{random},E'}(\boldsymbol{x})) \|_1 \right],$$
(9)

where we extend the $d$-dimension vector $\boldsymbol{I}_{\text{finetune},e}(\boldsymbol{x})$ to into a $2d$-dimension vector $\tilde{\boldsymbol{I}}_{\text{finetune},e}(\boldsymbol{x}) = [(\boldsymbol{I}^+_{\text{finetune},e}(\boldsymbol{x}))^{\mathrm{T}}, (-\boldsymbol{I}^-_{\text{finetune},e}(\boldsymbol{x}))^{\mathrm{T}}]^{\mathrm{T}} = [\max(\boldsymbol{I}_{\text{finetune},e}(\boldsymbol{x}), 0)^{\mathrm{T}}, -\min(\boldsymbol{I}_{\text{finetune},e}(\boldsymbol{x}), 0)^{\mathrm{T}}]^{\mathrm{T}} \in \mathbb{R}^{2d}$ without

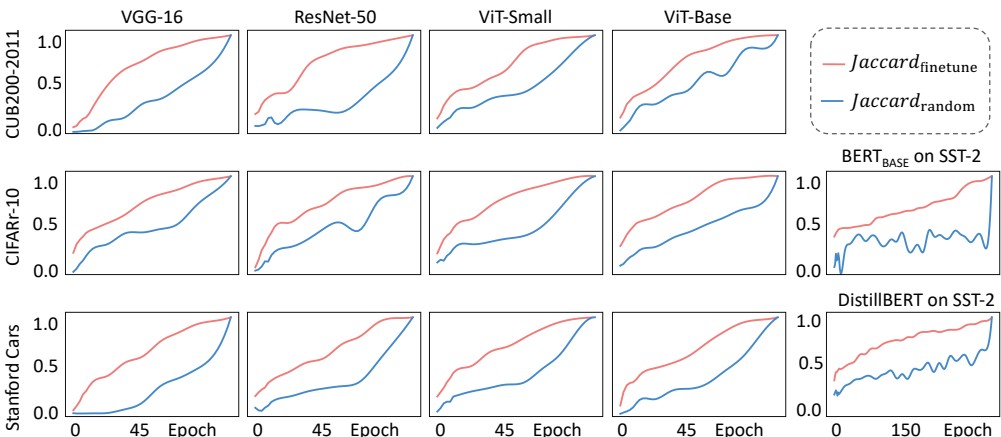

Figure 4: Changes of the Jaccard similarity $Jaccard_{\text{finetune}}$ and $Jaccard_{\text{finetune}}$ along with the epoch number. The similarity $Jaccard_{\text{finetune}}$ of the fine-tuned model exhibits a more sharp and stable increase with the epoch number than that of training from scratch $Jaccard_{\text{finetune}}$. This verifies the fine-tuned model learns target knowledge more quickly and directly, which accounts for its faster convergence.

negative elements. Accordingly, vectors $\tilde{I}_{\text{finetune},E}(\boldsymbol{x})$, $\tilde{I}_{\text{random},e'}(\boldsymbol{x})$, and $\tilde{I}_{\text{random},E'}(\boldsymbol{x})$ are constructed on $\boldsymbol{I}_{\text{finetune},E}(\boldsymbol{x})$, $\boldsymbol{I}_{\text{random},e'}(\boldsymbol{x})$, and $\boldsymbol{I}_{\text{random},E'}(\boldsymbol{x})$ to contain non-negative elements, respectively. Thus, a sharp increase of the similarity at early epochs indicates that the DNN encodes target knowledge quickly. Besides, a stable increase of the similarity along the epoch number, without significant fluctuations, demonstrates that the DNN encodes target knowledge directly.

**Experiments.** We conducted experiments to examine whether pre-training guided the fine-tuned model to encode target knowledge more quickly and directly than training from scratch. To this end, we employed fine-tuned DNNs and DNNs training from scratch introduced in the **experiment** paragraph of section 3.2.2 for evaluation. Fig. 4 reports the change of the similarity $Jaccard_{\text{finetune}}$ and $Jaccard_{\text{random}}$ along with the epoch number. We discovered that pre-training exhibited similar effects on guiding the fine-tuned model to learn target knowledge across different network architectures and datasets, as follows.

• Fig. 4 shows that the similarity $Jaccard_{\text{finetune}}$ first increased sharply in early epochs, then rose gradually and eventually saturated in later epochs, while the similarity $Jaccard_{\text{random}}$ usually exhibited the opposite trend, *i.e.,* first increasing gradually and then increasing rapidly in later epochs. This indicated that *pre-training enabled the fine-tuned model to learn target knowledge more quickly.*

• Fig. 4 also illustrates that the similarity $Jaccard_{\text{finetune}}$ usually increased stably along with the epoch number without significant fluctuations, while the similarity $Jaccard_{\text{random}}$ increased with ups and downs. This demonstrated that *pre-training guided the fine-tuned model to straightforwardly learned target knowledge, while the DNN training from scratch temporarily learned various knowledge and discarded task-irrelevant one later.*

## 4 CONCLUSION AND DISCUSSION

In this paper, we present an in-depth analysis to explain the benefits of pre-training, including the boosted accuracy and the accelerated convergence, from a game-theoretic view. To this end, we use interactions to explicitly quantify the knowledge encoded by the pre-trained model, and further analyze the effects of such knowledge on the downstream task, where the faithfulness of treating interactions as essential knowledge encoded by the DNN for inference has been theoretically ensured by a set of properties of interactions. We discover that compared to training from scratch, pre-training enables the fine-tuned model to encode more exclusively-learned and discriminative knowledge for inference, and to learn target knowledge more quickly and directly, which accounts for the superior classification performance and faster convergence of the fine-tuned model. This provides new insights into understanding pre-training, and may also guide new interesting directions on the fine-tuning behavior of the DNN for future studies.

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

# A    FAITHFULNESS OF USING INTERACTION PRIMITIVES TO REPRESENT KNOWLEDGE IN DNNS

Although there exist various ways to define/quantify the knowledge encoded by DNNs, a series of studies have theoretically proven and empirically verified the following properties as convincing evidence to take interactions as essential knowledge encoded by the DNN for inference.

(1) The **universal-matching property** in Theorem 3.1 and the **sparsity property** in equation 3 have mathematically guaranteed that a few interactions with salient effect $I(S|\boldsymbol{x})$ can faithfully explain the output of DNNs (Ren et al., 2024). Exactly speaking, given an arbitrary input sample with $n$ input variables, network outputs on $2^n$ differently masked samples can always be well approximated by a small set of salient interactions, no matter how we randomly mask this input sample.

(2) Li & Zhang (2023) have experimentally verified the **transferability property** and the **discriminative property** of interactions. Specfically, they have discovered that interactions exhibit considerable transferability across samples and across models, *i.e.,* interactions extracted from different samples in the same category are often similar, and different DNNs trained for the same task usually learns similar sets of interactions. They have also observed that a salient interaction has remarkable discrimination power in classification tasks, *i.e.,* the same salient interaction extracted from different samples usually pushes the DNN towards the classification of the same category.

(3) Ren et al. (2023a) have proven that interactions satisfy *efficiency, linearity, dummy, symmetry, anonymity, recursive, interaction distribution properties*, as follows.

① *Efficiency property.* The network output of a well-trained model $v(\boldsymbol{x})$ can be disentangled into the numerical effects of different interactions $v(\boldsymbol{x}) = \sum_{S \subseteq N} I(S|\boldsymbol{x})$.

② *Linearity property.* If the network output of the model $w$ is computed as the sum of the network output of the model $u$ and the network output of the model $v$, *i.e.,* $\forall S \subseteq N, w(\boldsymbol{x}_S) = u(\boldsymbol{x}_S) + v(\boldsymbol{x}_S)$, then the interaction effect of $S$ on the model $w$ can be computed as the sum of the interaction effect of $S$ on the model $u$ and that on the model $v$, $\forall S \subseteq N, I(S|\boldsymbol{x}) = I(S|\boldsymbol{x}) + I(S|\boldsymbol{x})$.

③ *Dummy property.* If the input variable $i$ is a dummy variable, *i.e.,* $\forall S \subseteq N \setminus \{i\}, v(\boldsymbol{x}_{S\cup\{i\}}) = v(\boldsymbol{x}_S) + v(\boldsymbol{x}_{\{i\}})$, then the input variable $i$ has no interaction with other input variables, $\forall S \subseteq N \setminus \{i\}, I(S \cup \{i\}|\boldsymbol{x}) = 0$.

④ *Symmetry property.* If input variables $i, j \in N$ cooperate with other input variables in $S \subseteq N \setminus \{i, j\}$ in the same way, $\forall S \subseteq N \setminus \{i, j\}, v(\boldsymbol{x}_{S\cup\{i\}}) = v(\boldsymbol{x}_{S\cup\{j\}})$, then input variables $i$ and $j$ have the same interaction effects, $\forall S \subseteq N \setminus \{i, j\}, I(S \cup \{i\}|\boldsymbol{x}) = I(S \cup \{j\}|\boldsymbol{x})$.

⑤ *Anonymity property.* For any permutations $\pi$ on $N$, then $\forall S \subseteq N, I(S|\boldsymbol{x}, v) = I(\pi S|\boldsymbol{x}, \pi v)$ is always guaranteed, where the new set of input variables $\pi S$ is defined as $\pi S = \{\pi(i), i \in S\}$, the new model $\pi v$ is defined as $(\pi v)(\boldsymbol{x}_{\pi S}) = v(\boldsymbol{x}_S)$. This suggests that permutation does not change the interaction effect.

⑥ *Recursive property.* The interaction effects can be calculated in a recursive manner. For $\forall i \in N, S \subseteq N\setminus\{i\}$, the interaction effect of $S \cup \{i\}$ can be computed as the difference between the interaction effect of $S$ with the presence of the variable $i$ and the interaction effect of $S$ with the absence of the variable $i$. That is, $\forall i \in N, S \subseteq N\setminus\{i\}, I(S \cup \{i\}|\boldsymbol{x}) = I(S|i \text{ is consistently present}, \boldsymbol{x}) - I(S|\boldsymbol{x})$, where $I(S|i \text{ is consistently present}, \boldsymbol{x}) = \sum_{L \subseteq S}(-1)^{|S|-|L|}v(\boldsymbol{x}_{L\cup\{i\}})$.

⑦ *Interaction distribution property.* This property describes how interactions are distributed for "interaction functions" (Sundararajan et al., 2020). An interaction function $v_T$ parameterized by a context $T$ is defined as follows. $\forall S \subseteq N$, if $T \subseteq S$, then $v_T(\boldsymbol{x}_S) = c$; Otherwise, $v_T(\boldsymbol{x}_S) = 0$. Thus, the interaction effect for an interaction function $v_T$ can be measured as, $I(T|\boldsymbol{x}) = c$, and $\forall S \neq T, I(S|\boldsymbol{x}) = 0$.

Besides, recent works have used interactions to explain the representation capacity of DNNs, including the generalization power (Zhang et al., 2021; Yao et al., 2023; Zhou et al., 2024), adversarial robustness (Ren et al., 2021), adversarial transferability (Wang et al., 2021), the learning difficulty of interactions (Liu et al., 2023; Ren et al., 2023b), and the representation bottleneck (Deng et al., 2022).

Thus, the above properties/usage of interactions ensure the faithfulness of taking the interaction as the essential knowledge encoded by the DNN for inference.

## B COMMON CONDITIONS FOR PROVING THE SPARSITY PROPERTY OF INTERACTIONS

Ren et al. (2024) have proven that under the following three common conditions, a well-trained DNN usually encodes a small set $\Omega_{\text{salient}}$ of salient interactions for inference, where $|\Omega_{\text{salient}}| \ll 2^n$.
(1) The DNN is assumed to not encode extremely high-order interactions, *i.e.,* high-order derivatives of the DNN output *w.r.t.* input variables are assumed to be zero
(2) The classification confidence of the DNN on partially masked input samples is assumed to monotonically increase with the size of the set of unmasked input variables.
(3) The network output of the masked input sample is assumed to neither be extremely high nor extremely low.

## C EXPERIMENTAL VERIFICATION OF HIGH CLASSIFICATION ACCURACY AND FAST CONVERGENCE SPEED OF THE FINE-TUNED MODEL

It has been widely acknowledged that the pre-training can help the fine-tuned model achieve better classification performance and converge faster than the DNN training from scratch (He et al., 2016; Devlin et al., 2019; Hendrycks et al., 2019). We experimentally verified the above two benefits brought by the pre-training, as follows.

Table 1 reports the classification accuracy of each pair of the fine-tuned model and the DNN training from scratch, which verified that the fine-tuned model usually achieved superior classification performance to the DNN training from scratch.

Fig. 5 shows the loss curves of each pair of the fine-tuned model and the DNN training from scratch, which verified that the fine-tuned model converged faster than the DNN training from scratch.

Table 1: Classification accuracy of each pair of the fine-tuned model and the DNN training from scratch. The fine-tuned model usually achieves superior classification performance to the DNN training from scratch.

| Dataset | Model architecture | Training from scratch | Fine-tuning |
|---------|-------------------|----------------------|-------------|
| CUB | VGG-16 | 23.5% | 71.2% |
| | ResNet-50 | 41.0% | 79.3% |
| | Vit-Small | 13.2% | 81.1% |
| | Vit-Base | 13.0% | 84.1% |
| Stanford Cars | VGG-16 | 18.7% | 78.3% |
| | ResNet-50 | 39.2% | 87.0 % |
| | Vit-Small | 7.7 % | 87.4% |
| | Vit-Base | 9.5% | 89.6% |
| CIFAR-10 | VGG-16 | 83.4% | 94.2% |
| | ResNet-50 | 83.2 % | 90.1% |
| | Vit-Small | 74.8% | 98.0% |
| | Vit-Base | 69.9% | 98.6% |
| SST-2 | $BERT_{BASE}$ | 79.1% | 91.5% |
| | DistillBERT | 78.5% | 89.1% |

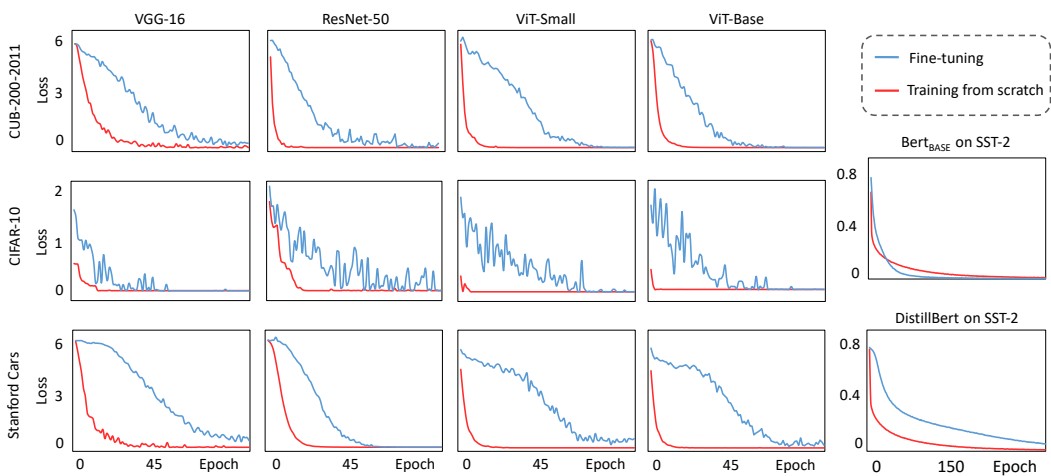

Figure 5: Loss curves of each pair of the fine-tuned model and the DNN training from scratch. The fine-tuned model converges faster than the DNN training from scratch.

# D  EXPERIMENTAL DETAILS

## D.1  ANNOTATING SEMANTICS PARTS

We follow (Li & Zhang, 2023; Ren et al., 2023a) to annotate semantic parts. Specifically, given an input sample $x \in \mathbb{R}^n$, the DNN theoretically encodes $2^n$ interactions. Thus, if the number of input variables $n$ is large enough, then the computational cost for calculating salient interactions is extremely high. To this end, we follow (Li & Zhang, 2023; Ren et al., 2023a) to annotate $10-12$ semantic parts in each input sample to reduce the computation burden, which also makes the annotated semantic parts are aligned over different samples in the same dataset. In this way, we take each semantic part of each input sample as a "single" input variable to the DNN.

For the SST-2 dataset, we followed settings in (Ren et al., 2023a) to select $50$ different sentences containing $10$ words with clear semantics to calculate interactions. Specifically, for each sentence, we took each word as an input variable, and obtained totally $n = 10$ variables.

For the CIFAR-10 dataset, we randomly selected 2 images for each category to annotate semantic parts to calculate interaction. followed settings in (Ren et al., 2023a) to for randomly selected images. Specifically, given an image, we first resized it to $224 \times 224$ before feeding it into the pre-trained model, and then divided the resized image into small patches of size $28 \times 28$, thereby obtaining $8 \times 8$ image patches in total. Considering the DNN mainly used foreground information/knowledge to make inference, we randomly selected $n = 12$ patches from $6 \times 6$ image patches located in the center of the image to reduce the computational cost.

For the CUB200-2011 dataset, we randomly selected 2 images for each category to annotate semantic parts and calculate interaction. Specifically, given an image, we divided the whole image into small patches of size $28 \times 28$, thereby obtaining $8 \times 8$ image patches in total. Similar to the settings in (Li & Zhang, 2023; Ren et al., 2023a) to annotate semantic parts for the CIFAR-10 dataset, we randomly selected $n = 12$ patches from $6 \times 6$ image patches located in the center of the image to calculate interactions, because the DNN mainly employed foreground information/knowledge for inference.

For the Stanford Cars dataset, we randomly selected 2 images for each category to annotate semantic parts and compute interactions. Specifically, given an image, we divided the whole image into small patches of size $28 \times 28$, thereby obtaining $8 \times 8$ image patches in total. Similar to the settings in (Li & Zhang, 2023; Ren et al., 2023a) to annotate semantic parts for the CIFAR-10 dataset, we randomly selected $n = 12$ patches from $6 \times 6$ image patches located in the center of the image to calculate interactions, because the DNN mainly employed foreground information/knowledge for inference.

## D.2 Details for Training Linear Classifier in Section 3.2.1

To extract pre-trained model's knowledge that is used for the downstream task, we employ a typical method, linear probing method (Alain & Bengio, 2016; Tenney et al., 2019; Chen et al., 2024). Specifically, given an input sample $x$ and a pre-trained model, let us fine-tune it on a certain downstream classification task and obtain the corresponding fine-tuned model. We freeze all network parameters in the pre-trained model, and use the feature $f(x)$ of its penultimate layer (*i.e.,* the layer preceding the classifier) to train a new linear classifier $W^T f(x) + b$ for the same downstream task.

In experiments, we set hyper-parameters to train the linear classifier the same as those to fine-tine the pre-trained model for fair comparisons. Specifically, we employed off-the-shelf VGG-16, ResNet-50, ViT-Small, and ViT-Base pre-trained on the ImageNet-1K dataset, and extracted the feature of the penultimate layer of each pre-trained model to train a linear classifier on the CUB200-2011, CIFAR-10, and Stanford Cars datasets for image classification, respectively. Each linear classifier was trained for 90 epochs using SGD with the momentum 0.9, weight decay $5 \times 10^{-4}$, and learning rate 0.01.

Besides, we also utilized off-the-shelf BERT$_{\text{BASE}}$ and DistillBERT models, and extracted the feature of the penultimate layer of each pre-trained model to train a linear classifier on the SST-2 dataset for binary sentiment classification, respectively. Each linear classifier was trained for 300 epochs with the learning rate $2e-5$.

## D.3 Details for Fine-tuning pre-trained models and Training DNN from scratch in Section 3.2.2

To enable fair comparisons, we set the model architecture of the DNN training from scratch the same as that of the fine-tuned model. Specifically, we fine-tuned the pre-trained VGG-16, ResNet-50, ViT-Small, and ViT-Base models on the CUB200-2011, CIFAR-10, and Stanford Cars datasets for 90 epochs using SGD with the momentum 0.9, weight decay $5 \times 10^{-4}$, and learning rate 0.01 for image classification, respectively. Correspondingly, we trained randomly initialized versions of the same models (VGG-16, ResNet-50, ViT-Small, and ViT-Base) on the same datasets for 90 epochs with the learning rate 0.1.

Besides, we fine-tuned the pre-trained BERT$_{\text{BASE}}$ and DistillBERT models on the SST-2 dataset for 300 epochs with the learning rate $2e-5$ for binary sentiment classification, respectively. Correspondingly, we trained randomly initialized versions of BERT$_{\text{BASE}}$ and DistillBERT models on the same dataset for 300 epochs with the learning rate $2e-4$.

The classification accuracy of each pair of the fine-tuned model and the DNN training from scratch was reported in Appendix C.

