# OpenReview forum: "Why pre-training is beneficial for downstream classification tasks?"
_ICLR.cc/2025/Conference — ICLR 2025 Conference Withdrawn Submission_

### Official Review · Reviewer_w69n · 2024-11-01

**Soundness:** 3
**Presentation:** 2
**Contribution:** 3
**Rating:** 5
**Confidence:** 4

**Summary:**

This paper proposes a metric to measure the knowledge learned by classification models, which can help track how knowledge changes during pre-training and fine-tuning. Using this measurement, the authors analyze the behavior of classification models, and claim that: a small portion of knowledge gained in pre-training is retained in fine-tuning. This retained knowledge is important for downstream tasks and cannot be learned from scratch. Also, pre-training makes fine-tuning faster and more direct in learning the target knowledge.

**Strengths:**

Instead of measuring learning by classification performance, the authors use the interaction between elements within an image. This approach offers a more interpretable view of the learning process.

**Weaknesses:**

Some concepts in the paper are not clearly defined mathematically and lack theoretical or experimental support. For example, in Line 340, the authors mention that pre-trained high-order knowledge is not “discriminative” for downstream tasks. It would help to clarify this—such as by explaining if low interaction scores or incorrect predictions caused by the discarded pre-trained knowledge support this idea.

The authors should prove the relationship between the proposed metric and the model performance is causal.

**Questions:**

- Does the proposed method only applicable to single-label classification tasks?

- In Eq. (2), the meaning of the indicator $(-1)^{|S|-|T|}$ needs clarification. How should we interpret the impact of (-1) or (+1) when S contains many elements, like 10? Also, should $T \in S$ be written as $T \subset S$ or $T \in 2^S$?

- What is the approximation error when using the salient interaction? Is this salient interaction universally applicable, or does it depend on the testing distribution?

- Since image variables are defined as patches, there will naturally be correlations between different patches. How does the method address this correlation?

- Is it possible that the pre-trained model learns redundant interactions, where pruning some interactions still leaves it strong enough for downstream tasks?

- It would help to calculate the correlation between the preservation ratio and model performance to support the claim that “pre-training makes the ﬁne-tuned model encodes more exclusively-learned and discriminative knowledge than the model training from scratch for inference.”

- Figure 4 is unclear; if all lines converge to 1.0 at the same time, how does this support the faster convergence rate of fine-tuning? Additionally, how can we verify that this relationship is causal and not just correlational? How do we ensure that an improvement in the interaction score actually enhances model capacity, rather than some latent factor improving both measures?

---

### Official Review · Reviewer_ojZT · 2024-11-02

**Soundness:** 2
**Presentation:** 2
**Contribution:** 2
**Rating:** 3
**Confidence:** 3

**Summary:**

The paper proposes a novel game-theoretic approach to explain why pre-training is beneficial for downstream tasks by analyzing and quantifying the knowledge encoded in pre-trained models. They represent knowledge in neural networks through "interactions" between input variables. An interaction measures how different input variables work together to contribute to the network's output.

The method relies on the following assumptons:

Sparsity property: Well-trained DNNs encode only a small number of significant interactions
Universal-matching property: Network outputs can be well explained by these sparse interactions

The authors propose metrics to quantify three types of knowledge during fine-tuning:
Preserved knowledge: Knowledge from pre-trained model that is kept during fine-tuning
Discarded knowledge: Pre-trained knowledge that is eliminated during fine-tuning
Newly-learned knowledge: Additional knowledge acquired during fine-tuning

The method is validated across multiple architectures (VGG-16, ResNet-50, ViT variants, BERT models) and datasets (CUB200-2011, CIFAR-10, Stanford Cars, SST-2), showing consistent patterns in how pre-training benefits downstream tasks.

**Strengths:**

This paper provides a systematic way to analyze pre-training benefits. The framework moves the field beyond just observing that pre-training works to understanding how and why it works.

**Weaknesses:**

The paper's empirical verifications focus more on validating their hypotheses about how pre-training helps downstream tasks, rather than directly verifying if the interactions they find are meaningful. While the authors thoroughly verify their hypotheses about pre-training benefits, they rely more on theoretical justification and prior work for the validity of their interaction analysis method itself.

The motivation for why this procedure should work is difficult to follow. Much of the text uses the justification of a proven sparsity property, but there isn't a clear intuition of why this property should hold for DNNs. Some of it is discussed in Appendix A and B, but would be helpful to have toy experiments to justify both this property but also the method itself recovers the correct ground truth of interactions in a controlled environment.

The method appears extremely similar to the SHAP (SHapley Additive exPlanations) approach. Much of this is mentioned in the citations of the related work, but the main contribution of this work appears to be the application of the SHAP approach to study the influence of training on fine-tuning.

Minor things:

Please specify the pre-training datasets for the language models.

Grammar Issue, Line 100

**Questions:**

How are the patch sizes for the images determined? Would this not change the types of interactions that are able to be found? Furthermore, the spatial coverage of each patch is unstructured with respect to the image content, would this method find interactions that  are smaller than the patch size? It's unclear how this would be possible.

Could counterfactual experiments be performed to validate the interactions determined in Figure 2 (e.g. Preserved knowledge) are indeed correctly identified?

---

### Official Review · Reviewer_FPwZ · 2024-11-03

**Soundness:** 3
**Presentation:** 3
**Contribution:** 3
**Rating:** 5
**Confidence:** 4

**Summary:**

The paper proposes a novel game-theoretic approach to explain the benefits of pre-training on downstream classification tasks by studying the knowledge that is preserved, discarded, or newly acquired during the fine-tuning process. The authors introduce interaction metrics to assess knowledge encoded by a pre-trained model and examine how it impacts downstream performance and convergence speed. They show that pre-training enables fine-tuned models to retain discriminative knowledge, leading to superior classification accuracy and faster convergence compared to models trained from scratch.

**Strengths:**

1. The paper is well-written, presenting complex ideas clearly and intuitively.

2. The use of game theory and an 'interaction metric' to study pre-training effects is both novel and insightful.

3. Extensive experiments across various architectures validate the proposed methods, underscoring their effectiveness and applicability.

**Weaknesses:**

1. Lack of clear motivation: The paper lacks clear motivation for using a game-theoretic approach, as similar conclusions about pre-training benefits have been reached through alternative methods, such as feature space analysis (e.g., Deng et al., 2023). Additionally, it does not provide a discussion comparing the advantages or unique insights offered by the game-theoretic approach over existing methods, leaving its added value unclear.

2. Lack of Practical Utility and Actionable Recommendations: Although the paper suggests the advantages of using pretrained models for downstream classification, it doesn’t provide concrete recommendations for how practitioners could adjust pre-training or fine-tuning protocols based on this insight of preserved or newly added knowledge. For example, it remains unclear how to retain more useful information or optimize the number of fine-tuning epochs.

3. Computational Complexity and Scalability Concerns: The paper does not address whether these metrics are feasible for large-scale models or datasets, raising concerns about scalability for real-world datasets.

4. Neglect of Discarded Knowledge’s Role in Robustness and Generalization: The paper assumes that discarded knowledge is irrelevant but does not analyze if this information affects model robustness, generalization to out-of-distribution data, or resilience to adversarial attacks. Understanding this could provide a more nuanced view of pre-training benefits and clarify whether certain discarded knowledge could enhance model robustness.

5. Continual and Online Learning Analysis: The paper does not explore scenarios where data arrives incrementally, as in continual or online learning. Research such as [1] suggests that in these cases, training from scratch may outperform using a pre-trained model. Examining the impact of preserved and discarded knowledge in such settings would extend the paper’s relevance.

6.  Evaluation with self-supervised pretrained model: Self-supervised learning has shown strong generalizability for classification and could yield different knowledge preservation dynamics compared to supervised pre-training. Experiments comparing self-supervised and supervised pre-training could reveal how each approach affects preserved and discarded knowledge, especially for tasks with limited labeled data.

7. Comparison of CNNs vs. Transformers :
Since CNNs and transformers represent data differently, they may encode and preserve different types of knowledge. It will be interesting to explore further whether CNNs or transformers retain more useful knowledge for downstream tasks, which could inform architecture selection for different scenarios.

8. The approach to annotating semantic parts for calculating interactions in this paper appears to be limited by arbitrary selection, as they randomly choose 2 images for each category to annotate semantic parts and compute interactions. While this strategy reduces computational complexity, it might introduce potential biases and undermines the rigor of interaction measurement by possibly excluding critical semantic parts. This might weaken the reliability of the results, as the randomly chosen annotations may not accurately represent the model's true interactions with significant features. Please elaborate on this.

References: [1]  Ash, J. and Adams, R.P., 2020. On warm-starting neural network training. Advances in neural information processing systems, 33, pp.3884-3894.

**Questions:**

Please address the questions in the Weakness section.

---

### Official Review · Reviewer_K5xX · 2024-11-04

**Soundness:** 2
**Presentation:** 1
**Contribution:** 1
**Rating:** 3
**Confidence:** 5

**Summary:**

The authors explore how pre-training is beneficial for downstream tasks, addressing a longstanding fundamental question in transfer learning. They connect recent progress in explaining DNN behavior through game-theoretic interactions to track changes in learned knowledge during the fine-tuning process. The authors propose methods to identify preserved, discarded, and newly acquired knowledge by tracking these changes. Equipped with these results, the paper concludes that, compared to training from scratch, pre-training allows the fine-tuned model to encode more exclusively-learned and discriminative knowledge for inference and to acquire target knowledge more quickly and directly. This accounts for the superior classification performance and faster convergence of the fine-tuned model.

**Strengths:**

1. The paper investigates a longstanding question in the transfer learning area and applies recently proposed game-theoretic interaction analyses to track the behavior of DNNs during the fine-tuning process.
2. It proposes metrics to explicitly quantify the preserved and discarded knowledge in pre-trained models, as well as the preserved and newly learned knowledge after fine-tuning.

**Weaknesses:**

1. Despite ambitious claims in the title and abstract, the manuscript primarily applies existing techniques to transfer learning, leading only to widely recognized conclusions without offering new theoretical insights or surprising experimental results. The manuscript falls short of the ICLR acceptance standard.
2. Given that fine-tuning commonly suffers from catastrophic forgetting and that improvements often stem from leveraging common pre-trained knowledge, the presented experimental results are predictable and lack significance.
3. The empirical analysis uses a broad array of target pre-trained models, including VGG-16, ResNet-50, ViT-small, ViT-base, BERT-base, and DistillBERT. These models differ significantly in terms of parameters, architectures, and pre-training strategies, spanning CNNs and transformers with varying datasets. The conclusions drawn from these comparisons merely confirm that the proposed metrics show consistent trends across different models.
4. Following previous point, the paper would benefit from a more detailed analysis, such as comparing supervised versus unsupervised pre-trained ResNet-50 models on ImageNet. It should investigate how variations in epochs and other hyperparameters might affect behavior, and whether the proposed metrics can detect these differences.
5. The domain gap between the target domain and the pre-trained dataset is a critical factor in fine-tuning behavior, yet the choice of fine-tuning datasets in the study seems arbitrary. The paper should clarify how the proposed metrics handle behaviors influenced by transfer gaps.
6. The experimental setup requires refinement; for example, previous research [1] indicates that linear probing is more challenging than full fine-tuning (requiring a much larger learning rate). The study treats these settings as equivalent. Note that different learning rates and training durations could alter conclusions regarding knowledge forgetting or not.
7. The paper focuses on classification problems. But transfer learning applications extend beyond this scope. The relationship between pre-text tasks and downstream tasks is also a critical factor in fine-tuning behavior.  Can these metrics extend to border transfer scenarios?

[1] He, Kaiming, et al. "Momentum contrast for unsupervised visual representation learning." CVPR 2020.

**Questions:**

Please refer to the weakness for my questions.

---

### Official Review · Reviewer_8Nq8 · 2024-11-09

**Soundness:** 3
**Presentation:** 3
**Contribution:** 3
**Rating:** 6
**Confidence:** 3

**Summary:**

This study analyzes how pre-training a model helps the fine-tuning process through knowledge-based decomposition. The authors define knowledge as the intersection of certain concepts within the input and empirically identify that: (i) the fine-tuning process utilizes only a small fraction of the pre-trained model's knowledge, with substantial portions being discarded; (ii) fine-tuning incorporates a significant amount of new, especially complex, knowledge; (iii) pre-training enables the model to acquire certain knowledge during fine-tuning that cannot be learned when training from scratch; and (iv) pre-training accelerates the knowledge learning process during fine-tuning.

**Strengths:**

- The study covers various datasets (CUB, Stanford Cars, CIFAR-10, etc), models (both CNN and transformer) and different domains (vision and language), providing a complete analysis of the knowledge-based understanding of pre-training/fine-tuning paradigm.
- This work is logically organized and covers several interesting understandings, like why pre-training/fine-tuning framework generally achieves better performance and fast convergence than training from scratch.
- Key takeaways are highlighted throughout the manuscripts, making it easy to understand.

**Weaknesses:**

- For section 3.2.2, whether there is any knowledge (intersections) that can only be learned through training from scratch and whether this knowledge is beneficial for the final classification. Analyzing the ratios of preserved knowledge acquired solely from training from scratch may only partially explain why pre-training and fine-tuning achieve better performance. Demonstrating that there are limited intersections unique to training from scratch would further strengthen this analysis.
- Reporting the final accuracy of different models in Figure 3 would be beneficial to verify whether the ratios of preserved knowledge are related to the final performance.
- Will different types of pre-training, such as self-supervised or adversarial pre-training, exhibit different properties in how knowledge changes during fine-tuning?

**Questions:**

Please refer to the weaknesses part.

---

### Note · Authors · 2024-11-15

I have read and agree with the venue's withdrawal policy on behalf of myself and my co-authors.